# Phosphorylated Adapter RNA Export Protein Is Methylated at Lys 381 by an Methyltransferase-like 21C (METTL21C)

**DOI:** 10.3390/ijms25010145

**Published:** 2023-12-21

**Authors:** Meiling Ke, Xiaoke Yu, Yuanyuan Sun, Shuai Han, Ling Wang, Tao Zhang, Wenxian Zeng, Hongzhao Lu

**Affiliations:** 1School of Biological Science and Engineering, Shaanxi University of Technology, Hanzhong 723001, China; kemeiling565@163.com (M.K.); yxk02200059@163.com (X.Y.); 17809166162@163.com (Y.S.); hs15689607278@163.com (S.H.); wl@snut.edu.cn (L.W.); zhangtao780823@snut.edu.cn (T.Z.); 2Engineering Research Center of Quality Improvement and Safety Control of Qinba Special Meat Products, Universities of Shaanxi Province, Hanzhong 723001, China; 3Qinba State Key Laboratory of Biological Resources and Ecological Environment, Hanzhong 723001, China; 4Shaanxi Union Research Center of University and Enterprise for Zhenba Bacon, Hanzhong 723001, China

**Keywords:** METTL21C, DF-1 cells, lysine trimethylation, LC-MS/MS, cell proliferation

## Abstract

Methyltransferase-like 21C (METTL21C) is a member of the non-histone methyltransferase superfamily, which mainly mediates the methylation of lysine (Lys) residues. The main types of modification are Lys dimethylation and trimethylation. However, at present, most of the studies on METTL21C are focused on humans and mice, and there are few reports on poultry. Therefore, chicken embryo fibroblasts (DF-1) were selected as the object of study. To explore the function of chicken METTL21C (chMETTL21C) in the proliferation of DF-1 cells, flow cytometry and qPCR were used to detect the function of chicken METTL21C in the proliferation of DF-1 cells. The results showed that overexpression of METTL21C blocked the cell cycle in the G1max S phase, thus inhibiting cell proliferation. In addition, based on proteomic analysis, stable overexpression of METTL21C may inhibit the proliferation of DF-1 cells by mediating lysine trimethylation of proliferation-related proteins phosphorylated adapter RNA export protein (PHAX), nucleoside diphosphate kinases (NDPKs), eukaryotic transcription extension factor (eukaryotic translation elongation factor 1A,e EF1A), and inversin (Invs). Through immunoprecipitation (co-IP) and liquid chromatography-mass spectrometry (LC-MS/MS) analysis, METTL21C-mediated PHAX Lys-381 methylation was confirmed to be involved in the regulation of DF-1 cell proliferation. The results of this study provide a reference for analyzing the methylation function of METTL21C and the mechanism of regulating the growth and development of chicken cells.

## 1. Introduction

Protein posttranslational modification (PTM) of proteins includes acetylation, glycosylation, methylation, succinylation, ubiquitin, and other types [1]. Among them, protein methylation is catalyzed by methyltransferase, and the methyl donor S-adenosylmethionine transfers methyl to other substrate protein amino acid residues, thus regulating the stability and function of cells [2]. Based on the different substrates of proteins involved, there are two types of methylation: histone and non-histone [3,4,5]. At present, there are many studies on histone methylation, and methylation sites and functions are also relatively clear. However, with the development of protein mass spectrometry identification technology, an increasing number of non-histone methylation sites have been found. Non-histone methylation is also a form of PTM in cells, and the biological function of non-histone methylation has been gradually revealed. It includes participation in many cellular activities, such as RNA processing, transcriptional regulation, DNA repair and gene expression regulation [3,6]. A large number of studies have shown that protein lysine methyltransferase can monomethylate, dimethylate or trimethylate Lys residues of non-histone substrates [7]. Chuikov et al. found that methyltransferase Set9 mediates demethylation of p53 at Lys-372 and promotes its binding on chromatin, leading to cell cycle arrest and apoptosis [8]. Moreover, ChoiJ et al. found that MEF2D, a member of the myocyte enhancer factor 2, has Lys-267 site trimethylation that can inhibit its transcriptional activity and then affect the regulation of skeletal muscle differentiation [9].

Non-histone methyltransferase-like 21 (METTL21) is a newly discovered methylation enzyme that belongs to the METTL subfamily, which includes METTL21A, METTL21B, METTL21C and METTL21D. Members of the METTL21 family modify the methylation of non-histone lysine [5] and have the ability to combine with molecular chaperones. UrulangodiM et al. found that methylation of members of the METTL21 family affects protein-protein interactions and participates in the regulation of cell proliferation, differentiation, migration and metabolism [10,11,12]. METTL21A methylation of Hsc70 inhibits the interaction between Hsc70 and Parkinson’s-associated a-synaptic protein, while speculating that it is involved in the process of cell proliferation and differentiation [13]. The Lys-165 methylation of eEF1A is mediated by METTL21B. Functioning as an indispensable protein in vivo, eEF1A is critical not only in mRNA translation but also in various cellular processes, such as cytoskeleton formation, apoptosis, nuclear export, protein degradation, and virus reproduction [14]. This methylation, orchestrated by METTL21B, is proposed to intricately regulate eEF1A activity in a tissue-specific manner [15], subsequently influencing the cellular proliferation process [16]. METTL21D binds to the molecular chaperone Val casein-containing protein (p97/valosin-containing protein, p97/VCP) to methylate the Lys-315 site of p97, which affects the activity of ATP and the proliferation of muscle cells. When the expression of METTL21D is dysregulated, it will lead to neuromuscular diseases [17]. METTL21C can modify the Lys-943 site of AARS1 in mouse skeletal muscle by methylation and regulate protein synthesis in muscle tissue [18]. Thus, members of the METTL21 subfamily are involved in a variety of cellular biological processes as non-histone lysine methyltransferases, but the comprehensive elucidation and analysis of their functions is limited.

Previous studies have confirmed that METTL21C modifies Hspa8/HSC70 by trimethylation to increase its stability, thus activating HSC70-mediated molecular chaperone autophagy and further degradation of MEF2A and MEF2D to maintain the homeostasis of slow muscle fibers in skeletal muscle [19]. In a previous study by our group, it was found that the expression level of METTL21C in the leg muscle of Lueyang silky chickens increased significantly, and chMETTL21C methylated insulin-like growth factor 2 mRNA-binding protein 1 (IGF2BP1) to eliminate the inhibitory effect of IGF2BP1 on chicken myoblast proliferation and regulate the proliferation activity of chicken myoblasts [20]. METTL21C promoted the differentiation of chicken myoblasts and modified the Lys-561 site of HSC70 by methylation. Compared with the Hsc70 group, the myocyte protein level of Hsc70 (K561) decreased [21]. However, which substrate proteins and sites are methylated by METTL21C has not been systematically elucidated. In this study, we selected chicken embryo fibroblasts (DF-1) as the research material to construct DF-1 cell lines that stably overexpressed METTL21C. The effect of METTL21C on the proliferation of DF-1 cells was analyzed by identifying the substrate protein modified by METTL21C methylation, as well as the modification level and site. These results provide new candidate substrate proteins for exploring the effect of METTL21C methylation on cell proliferation and provide fundamental data for revealing the molecular mechanism by which METTL21C regulates the proliferation of chicken embryo fibroblasts.

## 2. Results

### 2.1. Construction of the DF-1 Cell Line Overexpressing METTL21C

We first analyzed the mRNA expression level of the METTL21C gene in the process of DF-1 cell proliferation. The mRNA expression level of METTL21C was significantly decreased, especially at 24–48 h of DF-1 cell proliferation (Figure 1A). For this reason, we overexpressed METTL21C in DF-1 cells. Through the identification of the existing METTL21C overexpression vector, it was found that restriction enzyme digestion was consistent with the expected fragment size and could be used for follow-up experiments (Figure 1B). After packaging with lentivirus, HEK293T cells were in good condition, accounting for approximately 80% of 24 h fluorescence and 90% of 48 h fluorescence. These data show that the transfection efficiency of lentivirus is higher (Figure 1C). It is worth noting that in the process of lentivirus infection of DF-1 cells, DF-1 cells showed more fluorescence on the third day in the control group, while obvious fluorescence appeared on the ninth day after METTL21C overexpression. We speculate that the reason may be that the insertion of the target gene fragment affects downstream fluorescent protein expression, thus reducing the fluorescence intensity and gene expression ability (Figure 1D). Next, we used Puro to screen positive cells, and the control group obtained positive cells in approximately 3 days, whereas 15 days were needed to obtain METTL21C-overexpressing cell lines. To identify whether the cell line was constructed successfully, the expression of Flag was analyzed by Western blot. As expected, the specific band at approximately 28 kDa was the same as that of the METTL21C protein (28 kDa) and was upregulated 90-fold after gray analysis (Figure 1E). This result strongly indicates that the DF-1 cell line stably overexpressing METTL21C was successfully constructed.

### 2.2. Overexpression of METTL21C Inhibited DF-1 Cell Proliferation

Does METTL21C overexpression affect the biological characteristics of DF-1 cells? Interestingly, we found that the morphology of cells was regular and long fusiform in the control group, while there was no obvious good growth in the overexpression group (Figure 2A). Then, the growth rate was measured by the CCK8 assay in the overexpression group. Compared with the control group, the OD450 nm value decreased significantly after 72 h in the METTL21C overexpression group, and the growth curve slowed down significantly (*p* < 0.05) (*p* < 0.01) (Figure 2B). In addition, the number of cells stained by EDU showed that after overexpression of METTL21C, the number of EDU-positive cells decreased by 2.29% compared with that of the control group (Figure 2C), indicating that overexpression of METTL21C inhibited the DNA replication activity of DF-1 cells (*p* < 0.05). To assess which phase of the cell cycle might be influenced by METTL21C, cell cycle progression was analyzed by flow cytometry. In the control group, the proportion of cells in the G0/G1 phase was 54.09%, the proportion of cells in the S phase was 24.3%, and the proportion of cells in the G2/M phase was 21.61%. After overexpression of METTL21C, the proportion of cells in the G0/G1 phase was 55.75%, the proportion of cells in the S phase was 20.96%, and the proportion of cells in the G2/M phase was 23.29%. Compared with the control group, the overexpression group exhibited a significant increase in the proportion of cells in the G0/G1 phase, accompanied by decrease in the proportion of cells in the S phase. (Figure 2D). It is suggested that stable overexpression of METTL21C can change the distribution proportion of each cycle of DF-1 cells and mainly cause G1/S phase arrest in the DF-1 cell cycle (Figure 2E). Previous studies have shown that cyclin D1 protein and cyclin E1 protein can promote cell proliferation, and these two proteins can be used as markers for the detection of the cell cycle [16]. Cyclin p21 (cyclin-dependent kinase (CDK) inhibitor p21) and cell cycle inhibitor p27 (CDK inhibitor 1B) are two important cell cycle regulators and are considered to be universal inhibitors of CDKs [22]. To explore the molecular mechanism of stable overexpression of METTL21C in the arrest of the DF-1 cell cycle in the S phase of G1, we then studied the functional significance of p21, p27, cyclin D1 and cyclin E1 in this process. We performed RT-qPCR verification of the expression of several genes. The results showed that after stable overexpression of METTL21C, the expression of p27 was significantly upregulated, the expression of cyclin D1 and cyclin E1 was significantly downregulated, and the expression of p21 was downregulated (Figure 2F). Early studies have proven that p21 and p27 proteins play an important role in cell regulation and inhibit cell proliferation [23]. Although the p21 protein trend is significantly downregulated, as the upstream regulator of CDKs, its downregulation trend may not have a direct effect on cell proliferation, while cyclin D1 and cyclin E1, as cell cycle regulatory proteins, have a significant effect on cell proliferation. Therefore, it is reasonable to speculate that stable overexpression of METTL21C may upregulate the expression of p27 protein and downregulate the expression of cyclin D1 protein and cyclin E1 protein to block the cell cycle of DF-1 cells in the G1/S phase and inhibit cell proliferation. However, the specific mechanism by which METTL21C regulates the proliferation of DF-1 cells needs to be further explored.

### 2.3. Dimethylation and Trimethylation Detection

As a member of the lysine methyltransferase family, which substrate proteins are methylated by METTL21C to exert its biological activity? We first analyzed the level of METTL21C-methylated substrate protein by using lysine dimethylation and trimethylation pan-antibodies in DF-1 cells overexpressing METTL21C. The level of dimethylation of lysine was unstable, whereas trimethylation level was significantly increased, with a notable increase in the number of bands observed, especially when the molecular weight of the protein was between 35 kDa and 75 kDa (Figure 3A). It suggested that METTL21C, functioning as a methyltransferase, significantly impacts lysine trimethylation. Furthermore, the substrate proteins and methylation sites of METTL21C methylated by lysine were analyzed by LC-MS/MS. The results showed that the average number of methylation sites per 100 AAs in all identified lysine trimethylation-modified proteins was 11.19% (Figure 3B), indicating that there are more non-histone proteins in DF-1 cells that can affect their biological activity by methylation. The evaluation of METTL21C methylation activity on hundreds of proteins led to the identification of 322 lysine trimethylated proteins, 397 modified peptides and 494 methylation sites in the experimental group and overexpressed METTL21C group. Among the 317 lysine-trimethylated proteins, there were 385 quantifiable methylated peptides and 476 quantifiable methylation sites (Figure 3C). Further statistics showed that there were two or more modification sites on 23.91% of the methylated proteins, of which there were as many as 31 modification sites on the eEF1A protein (Figure 3D). It is suggested that a large number of proteins have lysine methylation sites in DF-1 cells, indicating the impacts of METTL21C on the biological process of DF-1 cells through the mediation of protein methylation.

### 2.4. Omics Analysis of Differentially Expressed Methylated Proteins

Because there are multiple methylation sites on the protein and the downregulation trends are different at different sites, this is impossible to quantify at the modified protein level, so quantitative analysis was carried out at the modified peptide level. In order to better compare the significant differences in methylated peptides between the two groups, a volcanic map of methylated peptides in the overexpressed METTL21C group and control group was drawn according to the two factors of expression difference multiple (FC) and *p* value (^t^ test). The results showed that compared with the control group, five modified peptides were significantly downregulated in DF-1 cells after METTL21C overexpression, namely, eEF1A, GUF1, inversin, WARS and PPIP5K2. Among them, seven modified peptides were significantly upregulated: LMO7, ACTG1, PHAX, MAP2, TRIO, NDPKs and Rab GDP dissociation inhibitors (Figure 4A). ACTG1, eEF1A, GUF1 and WARS are structural proteins. After that, the domain of the differentially expressed methylated peptide was predicted by InterProScan 5 version software. It is worth noting that there are three differentially methylated peptides enriched in the Tu GTP binding domain of the elongation factor related to protein synthesis, two differentially methylated peptides enriched with the elongation factor Tu binding domain 2, and one enriched in the cytoskeleton-related myosin junction domain. There was one differentially methylated peptide enriched in the microtubule binding domain related to the cell cycle (Figure 4B). It is speculated that methylation may be closely related to cell cycle regulation, cell proliferation and cytoskeleton formation.

Next, we focused on the function of the proteins that differentially expressed methylated peptides by GO analysis. The results showed that the differentially expressed peptides in different comparison groups were enriched in multiple items, including 20 biological processes, such as cell processes, biological regulation and metabolic processes, 14 cell components, such as cell components and organelles, and 7 molecular functions, such as binding, catalytic activity and molecular function regulators (Figure 4C). KEGG pathway enrichment analysis showed that the differentially expressed proteins between the two groups were mainly enriched in oxidative phosphorylation, purine metabolism, aminoacyl-tRNA biosynthesis, nuclear cytoplasmic transport, adhesion junction and actin cytoskeleton. After further analysis of the enrichment results, it was found that after methylation, the pathways of adhesion junction, pyrimidine metabolism and MAPK were upregulated, while oxidative phosphorylation, autophagy and sphingolipid signaling were downregulated (Figure 4D,E). It is speculated that METTL21C is a negative regulatory factor in the process of DF-1 cell proliferation. After classifying the differentially expressed methylated peptides in the overexpressed METTL21C group and the control group by hierarchical clustering algorithm, our data show that compared with the control group, there were 5 proteins and 6 modification sites in the overexpressed group. They are site K20 of eEF1A, site K908 of inositol pentaphosphate kinase 2, site K72 of GTP binding extension factor, site K259 and site K280 of tryptophan-tRNA synthetase and site K805 of reverse protein. Compared with the control group, seven methylated peptides and nine modification sites were highly expressed in the overexpression group. These were site K2791 of nonspecific serine/threonine protein kinase, site K238 and site K18 of actin, site K565 of LMO7, site K269 of RabGDP dissociation inhibitor, site K1812 and site K1815 of microtubule binding protein, and site K381 of nucleoside diphosphate kinase and phosphorylated junction RNA output protein (Figure 4F). Taken together, our data indicated that the number of methylated peptides and modification sites detected after METTL21C overexpression were higher than those in the control group, and four of the twelve methylated proteins were closely related to cell proliferation and cytoskeleton formation, PHAX, NDPKs, eEF1A and inversin.

### 2.5. Effect of Trimethylation of PHAX (K381) Mediated by METTL21C on Cell Proliferation

Based on the results of methylated 4D-Labelfree quantitative proteomics, we selected PHAX as the substrate protein for methylation to further verify the reliability and modification effect of proteomics. The interaction between METTL21C and PHAX was verified by immunoprecipitation (Co-IP). pCD513B-FLAG-METTL21C and pCD513B-HA-PHAX were transfected into HEK293T cells. Further analysis revealed that METTL21C and PHAX were detected in proteins pulled down by HA and FLAG. The Co-IP results indicated that there was a protein-based interaction between METTL21C and PHAX (Figure 5A). The trimethylation site of PHAX at Lys-381 has been found in 4D-label quantitative proteomics. In order to further verify whether methylation occurred at this site, we substituted lysine 381 with alanine to generate a methylation-deficient PHAX (K381A). Similarly, METTL21C and PHAX (K381A) were verified by the Co-IP assay. As expected, the interaction between METTL21C and Hsc70 was interrupted (Figure 5B). Then, we used IP to verify the level of METTL21C-mediated PHAX/PHAX (K381A) trimethylation. The results showed that the level of trimethylation after methylation of PHAX mediated by METTL21C was significantly higher than that of PHAX alone. However, the trimethylation level of the PHAX lysine trimethylation site (K381) decreased after mutation (Figure 5C), which indicated that METTL21C mediated trimethylation at PHAX Lys-K381.

In order to further detect the proliferation effect of cells after METTL21C methylation-modified PHAX, MTT detection was performed in this study. The results showed that there was no significant difference between the methylation group and the non-methylation group in the first 48 h, but the cell proliferation rate decreased at 72 h (*p* < 0.05) and significantly decreased at 96 h (*p* < 0.01) compared with the nonmethylation group (Figure 5D). This indicates that METTL21C inhibits cell proliferation by methylating the Lys-381 site of PHAX.

## 3. Discussion

Previous studies have shown that METTL21C is a methylated protein highly expressed in skeletal muscle that plays an important role in regulating the growth and development of skeletal muscle and the formation of a muscle phenotype [21]. Currently, studies on the regulatory role of METTL21C in the growth and development of muscle cells predominantly focus on human and mouse cells, with limited reports on its effects in poultry cells. It is not clear whether METTL21C participates in the growth, proliferation and development of poultry cells through methylation modification. Chicken embryo fibroblast (DF-1) is the only bird cell line that is widely used in research on gene function and gene editing in birds [24]. Therefore, in this study, DF-1 was taken as the research object. First, the DF-1 cell line stably overexpressing METTL21C was constructed by lentivirus, and then the recombinant vector was transfected into DF-1 cells. After METTL21C was overexpressed in DF-1 cells, the effect of METTL21C on the proliferation of chicken embryo fibroblasts was further explored.

Previous studies have found that METTL21C can affect the differentiation of mouse myoblasts by regulating the expression of MEF2 and thus affecting the differentiation of mouse myoblasts [25,26,27], but whether METTL21C affects the proliferation of mouse myoblasts is not clear. It is worth noting that the proliferative activity of DF-1 cells decreased significantly after overexpression of METTL21C. In order to better understand the effect of METTL21C on cell proliferation, the percentage of cells in each stage of the cell cycle was detected. Surprisingly, the results showed that METTL21C blocked the cell cycle of DF-1 cells in the G1/S phase and inhibited their proliferation. The progression of the cell cycle from the G1 phase to the S phase is regulated by many factors, among which the P27 and P21 proteins play a key role in regulating the cell cycle. Downregulation or upregulation of these factors will lead to cell cycle arrest. Cyclin E1 protein is a cell cycle marker [28]. The overexpression of cyclin E shortens the G1/S phase transition and accelerates the cell cycle, thus promoting cell proliferation and tumorigenesis [29,30]. Cyclin D1 protein is a key protein that regulates cell proliferation in the G1 phase of the cell cycle. It plays an important role in the G1/S phase [31,32], while the CDKI–CDK–cyclin system plays a key role in the regulation of the GS phase [33]. As a member of the CDKI family, p27 can negatively regulate the cell cycle and inhibit the growth of tumor cells [34]. Therefore, we speculate that METTL21C affects the proliferation process of DF-1 cells by regulating the expression of cell cycle cyclin E1 protein, cyclin D1 protein and p27 protein.

In recent years, more and more studies have confirmed that non-histone methylation may also be a common posttranslational modification that regulates the function of proteins. At present, members of the METTL21 family have been confirmed to play an important role as methyltransferases in regulating protein homeostasis. Thus far, the regulatory role of METTL21C methylation has become increasingly clear. In a genome-wide association analysis (GWAS), METTL21C was identified as a multiple factor regulating muscle and bone growth [20], suggesting that METTL21C may have a certain effect on muscle and bone cell proliferation [35]. WiedersteinJL used staining with MYH7 antibody, β-galactosidase staining and MYH2 antibody. The study confirmed that METTL21C was expressed in slow MYH7-positive skeletal muscle fibers, so METTL21C was identified as a skeletal muscle-specific lysine methyltransferase [16]. It is reported that METTL21C has a certain regulatory effect on skeletal muscle, including METTL21C affecting myotube formation by regulating the expression level of Mef2C [25]. Hence, METTL21C is very important in the growth and development of skeletal muscle. However, whether METTL21C participates in the growth, proliferation and development of muscle cells through methylation has not been systematically revealed.

Our previous studies found that the insulin-like growth factor II mRNA binding protein family (IGF2BP) plays an important biological role in cell polarization, proliferation, migration and differentiation in chicken myoblasts [36]. IGF2BP1, a member of its family, inhibited the proliferation of chicken myoblasts and further verified that METTL21C eliminated the inhibitory effect of IGF2BP1 on myoblast proliferation by regulating the modification of IGF2BP1 protein [20]. These results indicate that METTL21C affects the proliferation of muscle cells by regulating the trimethylation of the IGF2BP1 protein. Nevertheless, as a methylated protein, METTL21C may modify multiple substrate proteins in cells. However, there are no reports on the number and type of METTL21C methylation-modified proteins. For this reason, the molecular mechanism by which METTL21C-mediated protein methylation regulates the growth and development of muscle cells still needs to be further studied.

In the present study, we found that the level of trimethylation was more obvious and stable than that of lysine dimethylation. It is concluded that overexpression of METTL21C has a more obvious effect on lysine trimethylation. Several studies have shown that posttranslational modification regulates protein activity, so exploring the qualitative and quantitative of modification sites is essential to understand the regulatory mechanisms of cell function [37]. This study found that there are multiple lysine methylation sites on more than a fifth of proteins after METTL21C overexpression, and the average number of methylation sites is 11%. It is reasonable to speculate that METTL21C methylation is a common posttranslational modification behavior of proteins. In fact, all proteins play their biological role by using specific domains in cells [38]. Domain prediction is of great significance for the study of key functional regions of proteins and their potential biological roles. In order to elucidate the potential function of METTL21C in cell proliferation, we predicted the protein domain of differentially expressed methylated peptides based on InterProScan 5 version software. It was found that there were three Tu GTP binding domains related to protein synthesis and one related to a microtubule binding protein. Studies have shown that the extension factor Tu GTP binding domain (EF-Tu) is responsible for delivering aminoacyl-tRNA (aa-tRNA) molecules to ribosomes and plays a central role in protein synthesis [39]. Microtubule binding proteins can regulate the stability of microtubules, and microtubules are a component of the cytoskeleton, which plays an important role in regulating cell division, cell movement, cell morphology and polarity, and organelle distribution. Therefore, microtubule binding proteins also play a regulatory role in cell life activities [40,41]. It is further speculated that the methylation of EF-Tu and microtubule binding protein may affect protein synthesis and cytoskeleton formation in DF-1 cells, thus affecting cell proliferation.

In addition, we found that four differentially expressed substrate proteins, PHAX, NDPKs, eEF1a and inversin, were associated with cell proliferation after overexpression of METTL21C [42,43]. Therefore, do PHAX, NDPKs, eEF1A and inversin play a separate role in DF-1 cells, or do they affect cells after methylation together? It has been reported that nuclear diphosphate kinases (NDPKs) are involved in many regulatory processes related to cell proliferation, development and differentiation and are essential for DNA–RNA synthesis, cell division, macromolecular metabolism and growth [44]. MunshiR et al. found that the association of actin with eEF1A can attenuate the interaction between eEF1A and aminoacyl-tRNA, subsequently impacting cell proliferation through the inhibition of RNA translation [45]. Moreover, the inversin protein is involved in the regulation of the cell cycle and indirectly regulates the process of cell proliferation [46,47]. To date, research on the function of PHAX has mainly focused on the nuclear output of U snRNA, while further work on the proliferation of DF-1 cells needs to be conducted. Therefore, in the present study, PHAX protein with a significant difference was selected as the substrate protein modified by METTL21C methylation.

As a non-histone methylation enzyme, METTL21C is involved in the modification of a variety of substrate proteins. VCP [24] and HSPA8 [48] can be methylated in muscle cells, and Mef2C [25], and Mef2D [9] can be methylated in mouse myoblasts. Our experimental results also show that METTL21C regulates cell proliferation in DF-1 cells. PHAX is a phosphorylated nuclear export protein. The methylation of its site may affect the nuclear export process of small RNA and then affect cell proliferation [49]. After METTL21C-mediated methylation of PHAX, the rate of cell proliferation slows. It is reasonable to speculate that METTL21C, a regulator of PHAX, may inhibit the proliferation of DF-1 cells by mediating the trimethylation of PHAX and regulating the nucleation of nuclear small RNA [50]; however, more studies are needed to confirm whether the trimethylation of PHAX affects its own phosphorylation level and regulates the output of nuclear small RNA.

## 4. Materials and Methods

### 4.1. Cell Culture

The human embryonic kidney cells (HEK293T) used in this study were purchased from Wuhan Sanying Biotechnology Co., Ltd. (Wuhan, China). HEK293T cells were maintained at 37 °C in a 5% CO_2_ humidified incubator and were grown in 1× DMEM (90 mL) + fetal bovine serum (10 mL) + penicillin-streptomycin (1 mL). DF-1 cells were cultured at 39 °C in a 100% air incubator.

### 4.2. Lentivirus Packaging

The recombinant overexpression vector (pCD513B-Flag-METTL21C), lentivirus pCD513B plasmid, psPAX2 plasmid, and pMD2.G plasmids were preserved in our laboratory.

When the convergence degree of HEK293T cells reached 60%, the basic medium (without FBS) was starved, the packaging plasmid psPAX2 and the envelope plasmid pMD2.G were mixed into the tubes of the target plasmids pCD513B and pCD513B-Flag-METTL21C, and the tubes were incubated for 5 min at room temperature, while eddies were added to fully mix and the tubes were incubated for 20 min at room temperature. Then, the two mixtures were added to HEK293T cells and cultured in a 37 °C and 5% CO_2_ incubator, and the virus concentrate and virus solution were added to the centrifuge tube at a volume ratio of 1:4, mixed upside down, and placed at 4 °C for 45 min at 8069 rpm for centrifugation for 45 min to remove the supernatant and retain the white precipitate. The lentivirus supernatant was concentrated with a 10% volume of PBS, resuspended and precipitated, and the concentrated virus solution was obtained. When the DF-1 cells reached a confluence of 60%, the virus solution was added, and polycondensed amine was added to promote virus infection.

### 4.3. CCK-8

The cells were inoculated at a density of 35% per well in a 96-well plate, and 5 multiple wells were used for each group. The cells were cultured for 12 h, 24 h, 36 h, 48 h, 60 h and 72 h. Cell viability was determined as described previously [51] and cell viability was detected with cell counting kit (CCK-8) at 12, 24, 36, 48, 60, 72 h.

### 4.4. Detection of Cell Proliferation Activity by EdU Incorporation

When the cell density reached 70%, 1 mL EDU medium was added at 39 °C for 4 h. The cells were fixed for 15 min with 4% paraformaldehyde at room temperature. Then, 300 μL of 0.3% Triton X-100 was added and placed at room temperature for 15 min. Then, 250 μL of Click reaction solution was added to avoid light for 30 min at room temperature. Finally, the nucleus was stained for 15 min by adding DAPI staining solution in the dark at room temperature. Fluorescence detection was carried out under an inverted fluorescence microscope.

### 4.5. Flow Cytometry

Cell cycle analysis was performed as described previously [51]. When the cell density reached 80%, PBS was used to wash the bottom of the dish, the cells were digested with trypsin, and the cell suspension was collected and centrifuged at 2000 rpm/min for 5 min. The cells were fixed overnight with 70% ethanol at 4 °C and then stained with RNase and propidium iodide for 30 min in the dark at 4 °C. The DNA content in each cell cycle was detected by flow cytometry to determine the proportion of cells in the G0/G1 phase, S phase and G2/M phase.

### 4.6. Real-Time Fluorescence Quantitative PCR

For cell resuscitation in a petri dish, when the cell density reached 80%, TRIzol was added to cover the bottom of the dish, and the dish was placed on ice. The cell lysate was collected into an RNase-free tube, and chloroform was added to shake violently and rest on ice. After centrifugation, isopropanol was added to the supernatant and incubated on ice. Then, the sample was centrifuged to remove the supernatant, the RNA precipitate was washed with 75% ethanol in advance, ethanol was removed by centrifugation, and DEPC water was added to dissolve the RNA when the RNA precipitate became translucent.

PCR was performed using the following primer sequences (Table 1) for detection of the METTL21C and β-actin transcripts: METTL21C-F 5′-GAAGAAGCAGCCTGTGAAGAGC, R 5′-TCAAAGTAGGGGAACTGGTGGG, β-actin-F 5′-CTGTGCCCATCTATGAAGGCTA, and R 5′-ATTTCTCTCTCGGCTGTGGTG.

### 4.7. Antibodies and Immunoblotting

Antibodies against the proteins β-actin (mouse), Flag (mouse), dimethylation and trimethylation were all purchased from Proteintech Company (Wuhan, China). The secondary anti-rabbit and anti-mouse immunoglobulin antibodies were used to detect the primary antibodies, and immunoreactivity was detected by an ECL Prime Western blotting detection system (GE Healthcare, Beijing, China).

### 4.8. Quantitative Proteomic Analysis of Methylated 4D-Label-Free

The cell precipitates of the experimental group (OE-METTL21C) and the control group (empty vector) were detected by LC-MS/MS with 3 parallel repeats in each group, and the collected data were identified by Peaks 11 version software for protein library identification and quantitative analysis. The Venn diagram of the experimental group and the control group was drawn by the OmicShare online tool, and the hierarchical clustering heatmap was generated by Complexheatmap R software (R Version 3.4). The subcellular localization of the protein was analyzed by CELLO (http://cello.life.nctu.edu.tw/ (accessed on 5 February 2023)) v.2.5 version software (URL accessed on 5 February 2023), the protein domain was predicted by InterProScan 5 version software, domain enrichment analysis was carried out by Fisher’s precise test (Fisher’s Exact Test), and GO annotation was carried out by Blast2GO (https://www.blast2go.com/ (accessed on 11 January 2023)) software (URL accessed on 11 January 2023). KAAS (KEGG Automatic Annotation Server) 3 April 2015 version software was used to annotate the database access of KEGG (Kyoto Encyclopedia of Genes and Genomes).

### 4.9. Construction of the PHAX Overexpression Vector

PHAX primers with HA tags were designed and digested with BamH I and EcoR I, ligated to the pCD513B empty vector, and ligated and transformed into DH5α competent cells. Appropriate amounts of competent cells were evenly coated on LB plates containing ampicillin and cultured at 37 °C. The clones were selected, colony PCR was performed, the positive clones were screened, and the plasmids were extracted and verified by restriction endonuclease digestion.

### 4.10. Co-IP

The overexpression and Co-IP assays were carried out in cells by transfecting pCD513B-FLAG-METTL21C and pCD513B-HA-PHAX/pCD513B-HA-PHAX (K381) into HEK293T cells using M5 Hiper Lipo2000 transfection reagent (Mei5 Biotechnology Co., Ltd., Beijing, China).

## 5. Conclusions

Our studies have shown that stable overexpression of METTL21C can inhibit the proliferation of DF-1 cells and that stable overexpression of METTL21C protein lysine trimethylation-modified peptides participates in cell adhesion and other biosynthetic pathways. Among them, the methylated proteins PHAX, NDPKs, eEF1A and inversin, showing significant differences, are all related to cell proliferation so as to explore the interaction between METTL21C and PHAX. The results showed that METTL21C is an important protein in the proliferation and development of chicken embryo fibroblasts, and PHAX Lys-381 may be a potential target by which METTL21C regulates the proliferation of chicken embryo fibroblasts.

## Figures and Tables

**Figure 1 ijms-25-00145-f001:**
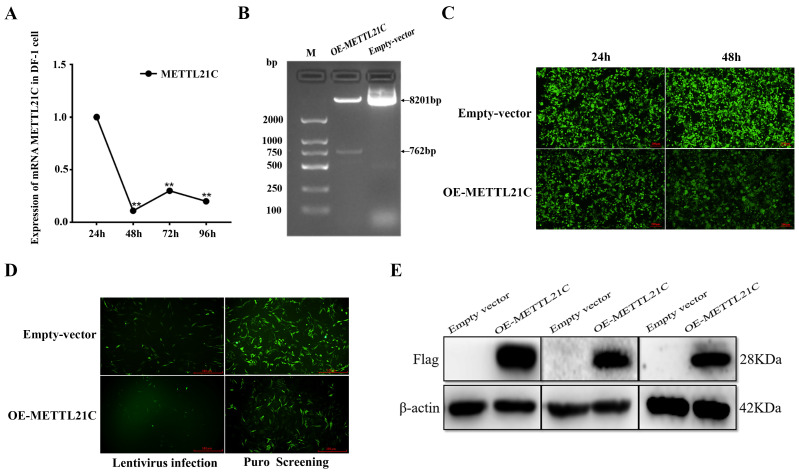
Construction of a DF-1 cell line overexpressing METTL21C. (**A**) Detection of METTL21C gene expression in DF-1 cells by qPCR. ** *p* < 0.01. (**B**) The recombinant plasmid was identified by restriction endonuclease (BamHI/EcoRI) product electrophoresis. (**C**) Packaging of lentivirus, scale bar is 1:200 μm. (**D**) Fluorescence observation of lentivirus infection in DF-1 cells and screening of positive cell lines by puromycin, scale bar is 1:100 μm. (**E**) Western blot detection of overexpression of METTL21C in DF-1 cells.

**Figure 2 ijms-25-00145-f002:**
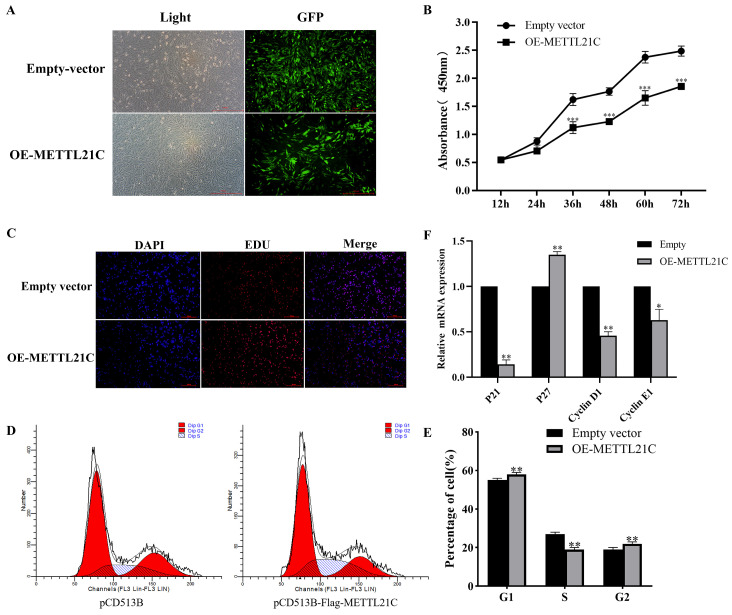
Overexpression of METTL21C slows cell cycle progression and inhibits cell proliferation in DF-1 cells. (**A**) Morphological changes in DF-1 cells were observed by fluorescence inverted microscopy, scale bar is 1:100 μm. (**B**) Proliferation of DF-1 cells was detected by the CCK-8 method at 12, 24, 36, 48, 60 and 72 h after cell inoculation. *** *p* < 0.001. (**C**) The rate of EDU-positive cells was analyzed by ImageJ, and the effect of overexpression of METTL21C on the proliferation activity of DF-1 cells was detected, scale bar is 1:100 μm. (**D**) Flow cytometry analysis of cell cycle progression in each phase of the cell cycle in the control group (empty vector) and experimental group (OE-METTL21C). (**E**) Quantification of cell distribution at each stage of the cell cycle, ** *p* < 0.01. (**F**) RT-qPCR was used to detect the cyclin protein and cell distribution was quantified at each stage of the cell cycle. * *p* < 0.05, ** *p* < 0.01.

**Figure 3 ijms-25-00145-f003:**
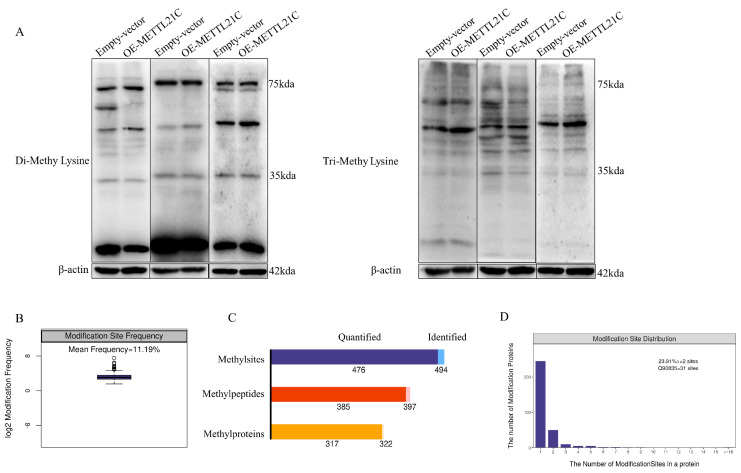
Lysine methylation level verification. (**A**) METTL21C-mediated dimethylation and trimethylation of lysine (detection of all METTL21C-methylated substrate proteins in DF-1 cells by lysine dimethyl and trimethylated pan-antibodies). (**B**) The distribution frequency of lysine methylation modification sites was analyzed by proteomics. (**C**) The distribution of methylation modification sites was analyzed by proteomics. (**D**) The frequencies of lysine methylation sites were analyzed with Peaks 11 version software.

**Figure 4 ijms-25-00145-f004:**
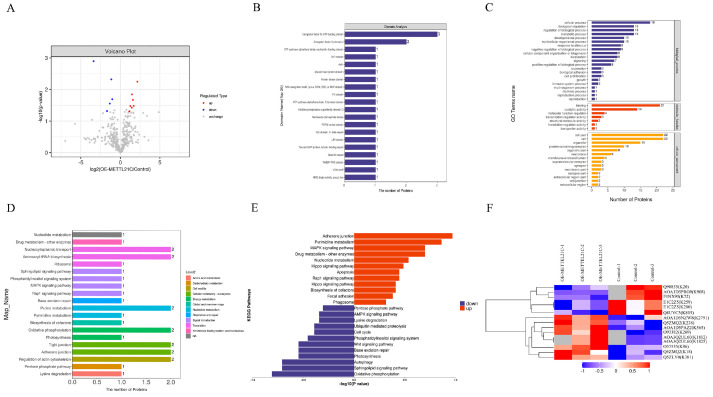
Differential expression of methylated protein analysis. (**A**) Proteome differentially expressed methylated peptide analysis. (**B**) Analysis of the protein domain to which the proteome differentially expressed modified peptide belongs. (**C**) GO annotation statistics of proteins to which proteome differentially expressed methylated peptide belongs. (**D**,**E**) Proteome differentially expressed methylated peptide belongs to protein KEGG pathway annotation. (**F**) Proteome differentially expressed methylated peptide cluster analysis.

**Figure 5 ijms-25-00145-f005:**
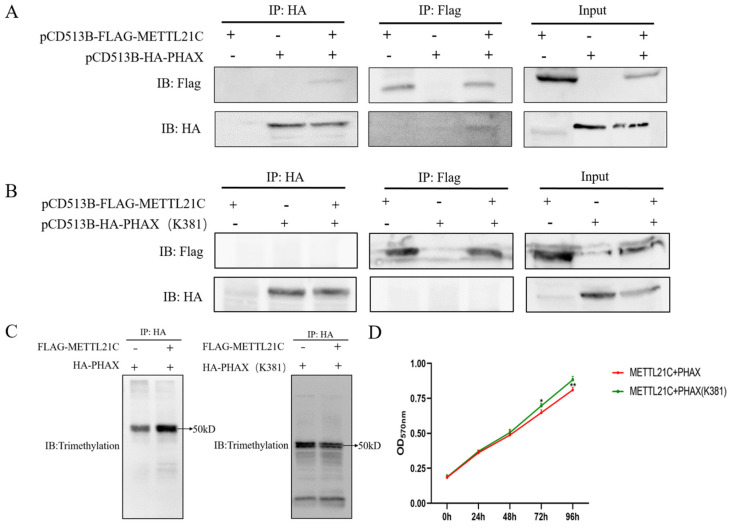
Analysis of the interaction between METTL21C and PHAX. (**A**) The interaction between METTL21C and PHAX was detected by CO-IP technology. (**B**) The interaction between METTL21C and PHAX (K381) was detected by CO-IP technology. (**C**) PHAX trimethylation level detection. (**D**) MTT detected the effect of overexpression of METTL21C and PHAX/PHAX (K381) on the proliferation of DF-1 cells.

**Table 1 ijms-25-00145-t001:** Primers for real-time qPCR.

Primers	Sequences (5′–3′)
METTL21C-F	GAAGAAGCAGCCTGTGAAGAGC
METTL21C-R	TCAAAGTAGGGGAACTGGTGGG
β-actin-F	CTGTGCCCATCTATGAAGGCTA
β-actin-R	ATTTCTCTCTCGGCTGTGGTG

## Data Availability

Data are contained within the article.

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
