# Peer review of "Phosphorylated Adapter RNA Export Protein Is Methylated at Lys 381 by an Methyltransferase-like 21C (METTL21C)"

_ijms, 2023, doi:10.3390/ijms25010145_

Round 1
Reviewer 1 Report
Comments and Suggestions for Authors
Without a doubt, this article is devoted to an important and interesting topic - experimental confirmation of the methylation activity of METTL21C - methyltransferase and the study of its effect on other proteins, including non-histone ones. An enormous amount of experimental work has been done. However, the article requires careful editing, since sometimes it is not clear what is methylated and who did it. So, the title says “Phosphorylated adapter RNA export protein is Methylated at 2 Lys 381 by a METTL21C Like Methyltransferase.”
What does the word "like" in the title refer to? Is the METTL21C protein a methyltransferase or is it just similar to one?
There are many such incomprehensible sentences in the text, for example, “Chuikov et al found that Set9 dimethylation modifies the Lys-372 site of the p53 protein in the nucleus, which promotes the stability of the binding part of the p53 protein to chromatin, leading to cell cycle arrest and apoptosis[8]." Or "In particular, METTL21C promotes the growth and development of muscle cells, but at present, related studies are mainly focused on human and mouse cells, but there are few reports on poultry cells."
In addition, there are typos, for example “According to the dif-40 ferent substrate proteins of methylation, protein methylation can be divided into two 41 forms: histone methylation and no-nhistone methylation[3-5].” and article format violations, such as lines 131-172.
It is not clear in the legend to Figure 3 which protein was used to assess the methylase activity of METTL21C. The legend should be expanded.
Figures 3 C and E are generally astonishing - it turns out that experiments assessing methylase activity were carried out on hundreds of proteins? This was left unexplained in the text of the article.
All figures captions should be improved and explained carefully. Please, add all nessesary details, methods and data used. Each figure caption should be understandable without reference to the main text of the manuscript
Comments on the Quality of English Language
I don't understand phrases like that "Omics analysis of lysine differentially expressed methylated proteins" - what do you mean? There are a lot of such phrases in the manuscript text. Please, ask somebody English-speaking to help you.
Author Response
Dear reviewer,
Thank you very much for your comments and professional advice. These opinions help to improve academic rigor of our article. Based on your suggestion and request, we have revised the original manuscript. Please see the attachment.

Reviewer 2 Report
Comments and Suggestions for Authors
The study investigates the role of Methyltransferase-like 21C (METTL21C) in the proliferation of chicken embryo fibroblasts (DF-1 cells). The authors highlight the limited research on METTL21C in poultry compared to humans and mice, which motivated their study. This work has some interest, but the authors should conduct various modifications before it can be accepted for publication.
The authors should check the instructions to verify the style for references, as they usually refer to the last name of the author and the first letter of his first name without spaces.
Figure 2D, 3D, 4 and 5D should be enlarged and their quality should be improved. Moreover, there are some non-latin characters in Figure 5C.
How does the expression of METTL21C in poultry cells compare to its expression in human and mouse cells? Are there any differences or similarities?
How does METTL21C influence the cell cycle progression of DF-1 cells, specifically in the G1/S phase? Are there any mechanistic insights into this process?
Can the observed effect of METTL21C on DF-1 cell proliferation be attributed solely to its regulation of CyclinE1, CyclinD1, and p27 protein expression? Are there any additional factors or pathways involved?
How does the level of trimethylation compare to lysine dimethylation upon METTL21C overexpression? What are the implications of this observation for the regulation of lysine trimethylation by METTL21C?
What is the mechanism by which the methylation of eEF1A affects cell proliferation? How does the methylation of Inversin regulate the cell cycle and indirectly impact cell proliferation?
In Materials and Methods section, an appropriate schematic should be added in order to underline the different steps of the present research.
Author Response

(The authors gave the same response as above.)

Round 2
Reviewer 2 Report
Comments and Suggestions for Authors
The authors have performed all the necessary modifications to their manuscript. Thus, it can be accepted for publication.